# Jointly Learning "What" and "How" from Instructions and Goal-States

**Dzmitry Bahdanau**[*]
MILA, Universite de Montreal
bahdanau@iro.umontreal.ca

**Felix Hill, Jan Leike, Edward Hughes, Pushmeet Kohli, Edward Grefenstette**
DeepMind, London

## Abstract

Training agents to follow instructions requires some way of rewarding them for behavior which accomplishes the intent of the instruction. For non-trivial instructions, which may be either underspecified or contain some ambiguity, it can be difficult or impossible to specify a reward function or obtain relatable expert trajectories for the agent to imitate. For these scenarios, we introduce a method which requires only pairs of instructions and examples of goal states, from which we can jointly learn a model of the instruction-conditional reward and a policy which executes instructions. Our experiments in a gridworld compare the effectiveness of our method with that of RL in a control setting with available ground-truth reward. We furthermore evaluate the generalization of our approach to unseen instructions, and to scenarios where environment dynamics change outside of training, requiring fine-tuning of the policy "in the wild".

## 1 Introduction

One of the many goals of artificial intelligence is to build agents that are able to follow arbitrary instructions given by humans. To specify what an instruction-following agent needs to do, researchers typically rely on either reinforcement learning (RL) or on expert demonstrations. In the RL approach (e.g. Branavan et al. (2009); Hermann et al. (2017); Chaplot et al. (2017)) the agent is rewarded for executing the instruction correctly, and must understand the task only from the rewards it receives. In methods that rely on demonstrations, often referred to as imitation learning (IL; Chen and Mooney (2011); Artzi and Zettlemoyer (2013); Andreas and Klein (2015); Mei et al. (2016)), an agent is trained to reproduce the behaviour of the expert when given the same instruction in the same initial state. Both the RL and IL approaches to training instruction-following agents suffer from certain limitations: a formal reward specification is required for RL and complete traces with the agent's actions are required for IL (and the agent's action space might be different from the human's).

We explore an alternative approach that avoids both the idiosyncrasies of human-provided trajectories and the burden of reward programming. We focus on the case where instructions implicitly characterize a set of goal-states (e.g. "Move things around so that a red rectangle is north of a green sphere.") and where the agent's task is to bring the world in one such state. Given a dataset of instructions and respective goal-state examples provided by an expert, we train a discriminator network and a policy network that focus on the "what to do" and "how to do it" aspects of the tasks, respectively. We call our approach Adversarial Goal-Induced Learning from Examples (AGILE). AGILE is strongly inspired by Inverse Reinforcement Learning (IRL; Ng and Russell (2000); Ziebart et al. (2008)) methods in general and Generative Adversarial Imitation Learning (Ho and Ermon, 2016) in particular, but it differs in that the policy and the discriminator are conditioned on an instruction and that goal-states instead of complete trajectories are given to the discriminator. A unique property of AGILE is that the "what to do" knowledge of the discriminator can be reused to improve the agent's performance on novel instructions or let the agent adapt to a change of the environment.

---

[*]Most of the work was done during an internship at DeepMind, London.

To validate the approach, we experiment with verifiable structured instructions in a gridworld. We show that learning speed and performance of AGILE are comparable to those of vanilla RL, as well as investigate the data efficiency (in terms of the number of positive examples) of AGILE. We then demonstrate that AGILE affords greater and more flexible generalisation than vanilla RL.

## 2 ADVERSARIAL GOAL-INDUCED LEARNING FROM EXAMPLES

We consider a setup in which an agent is is given a dataset $\mathcal{D}$ of $(c_i, g_i)$ pairs, where $c_i$ is an instruction and $g_i$ is an example goal-state for $c_i$. An example $c_i$ could be "build a tower from blocks on the table" and the respective $g_i$ could be a visual observation from which it is clear that blocks form a tower. We furthermore assume that the agent is provided with a stream of training instances $\mathcal{G}$. Each training instance constitutes an instruction $c$ and an initial state $s_0$ from which the instruction can be successfully executed. Our algorithm trains a policy $\pi_\theta$ with parameters $\theta$ on the instances from $\mathcal{G}$ using only examples from $\mathcal{D}$ to understand the semantics of the instructions. In order to train our agent without an explicit reward but by using just examples from $\mathcal{D}$ we introduce an additional network $D_\phi$ which we call the discriminator whose purpose is to define a meaningful reward function for training $\pi_\theta$. Specifically, the discriminator $D_\phi$ is trained to predict whether a state $s$ is a goal-state for an instruction $c$. The discriminator's positive examples are fetched from $\mathcal{D}$, whereas its negative examples come from the agent's attempts to solve the training instances from $\mathcal{G}$. Formally speaking, the policy is trained to maximize a return $R_\pi(\theta)$ and the discriminator is trained to minimize a cross-entropy loss $L_D(\phi)$, the equations for which are listed below:

$$R_\pi(\theta) = \mathop{\mathbb{E}}_{(c, s_{1:\infty}) \sim \mathcal{G}^{\pi_\theta}} \sum_{t=1}^{\infty} \gamma^{t-1} \left[ D_\phi(c, s_t) > 0.5 \right] + \alpha H(\pi_\theta), \tag{1}$$

$$L_D(\phi) = \mathop{\mathbb{E}}_{(c, s) \sim B} - \log(1 - D_\phi(c, s)) + \mathop{\mathbb{E}}_{(c_i, g_i) \sim \mathcal{D}} - \log D_\phi(c_i, g_i). \tag{2}$$

In the equations above square brackets $[x]$ stand for the indicator function, i.e. $[x] = 1$ iff $x > 0$ and 0 otherwise. $\gamma$ is the discount factor. With $(c, s_{1:\infty}) \sim \mathcal{G}^{\pi_\theta}$ we denote a state trajectory that was obtained by sampling $(c, s_0) \sim \mathcal{G}$ and running $\pi_\theta$ conditioned on $c$ starting from $s_0$. $B$ denotes a replay buffer to which $(c, s)$ pairs from $T$-step episodes are added, or in other words, it is the undiscounted occupancy measure over the first $T$ steps. $D_\phi(c, s)$ refers to the probability of $(c, s)$ having a positive label according to the discriminator. $H(\pi_\theta)$ is the policy's entropy, and $\alpha$ is a hyperparameter. Figure 1 illustrates the complete approach.

## 3 SETUP

In order to investigate AGILE we built a minimalist $5 \times 5$ gridworld surrounded by walls. The cells of the grid can be occupied by blocks of 3 possible shapes and 3 possible colors. The grid also contains an agent sprite. The agent may carry a block; when it does so, the agent sprite changes color. When the agent is free, i.e. when it does not carry anything, it is able to enter cells with blocks. A free agent can pick a block in the cell where both are situated. An agent that carries a block can not enter non-empty cells, but it can instead drop the block that it carries in any empty cell. Both picking up and dropping are realized by the INTERACT action. Other available actions are LEFT, RIGHT, UP and DOWN and NOOP. We render the state of the world as an image (see Figure 1 for examples of rendered world states) and we use it as the input for all neural networks.

We defined a minimalistic task that requires the agent to understand 5 spatial relations (*NorthFrom*, *SouthFrom*, *EastFrom*, *WestFrom*, *SameLocation*) in the context of our gridworld. The operands of relations can be either the blocks, which are referred to by their shapes and colors, or the agent itself. The instructions are represented as programs in a simple functional programming language. A program can verify that a block is one step in a specified direction from another block. For example, the program *NorthForm(Color('red', Shape('circle', SCENE)), Color('blue', Shape('square', SCENE)))* returns *True* if and only if there exists a red circle right north from a blue square. Figure 1 shows two examples of instructions and their respective goal states.

## 4 EXPERIMENTS

We trained AGILE and RL agents on episodes of 30 steps. We considered an episode to be a success if the final state was a goal state as judged by the ground-truth program. We use the success rate (i.e. the ratio of successful episodes) as our main performance metric for the agents. More detailed presentation of our results can be found in Appendix A.

**AGILE vs RL**   In our first experiment we compare the performance of AGILE against a vanilla RL baseline. For the RL baseline we used the ground-truth goal checker to provide binary reward to the agent. While AGILE training was able to take off without any auxiliary tasks, we found it necessary to add an auxiliary task of reward prediction (RP; Jaderberg et al., 2016) to make the RL baseline work well. The best performance for AGILE was $0.95$, which is close to $0.99$ attained by RL-RP, albeit not perfect.

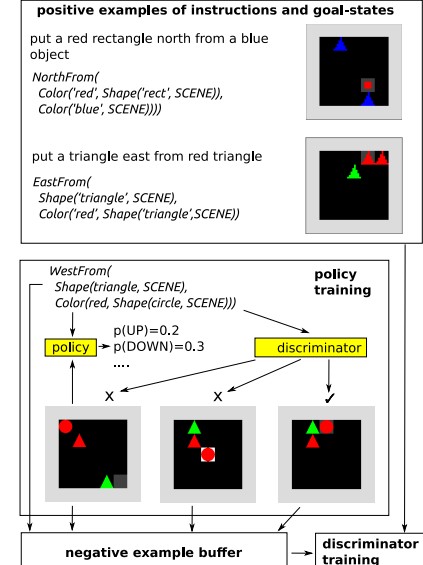

**Data Efficiency**   We measure how many examples of instructions and goal-states are required by AGILE in order to understand the semantics of the GridLU-Relations instruction language. The AGILE-trained agent succeeds in more than 50% of cases starting from $2^{13}$ examples, but as many as $2^{17}$ is required for the best performance.

**Generalization**   We test generalization of agents trained by AGILE to unseen instructions. We split instructions into a training and a test set by holding out instructions that contain subexpressions *Color('blue', ...)* and *Shape('sphere', ...)* (this includes instructions which refer to blue spheres but not only them). This split is applied to both the dataset $\mathcal{D}$ and the instance generator $\mathcal{G}$. To make the task harder, at test time we also add two additional objects to the gridworld, a nonblue circle and a blue non-circle. The success rate dropped from $0.88$ on the training set to $0.21$ when we evaluated the trained policy on the test set. To find out whether it is the policy's or the dis-

Figure 1: Top: examples of instructions and respective goal-states. Instructions are displayed in italic together with their natural language equivalents in a normal font. Bottom: schematic illustration of adversarial learning with goal-states examples.

criminator's failure to generalize, we fine-tuned the policy on the test instructions while freezing the discriminator's weights, which improved the average success rate to $0.50$, which suggest that the discriminator generalized better to the test situation.

To further investigate AGILE's potential uses, we performed an additional generalization experiment. We modified the physics of the game by making red square blocks immovable, simulating a scenario where low battery or damaged actuators render a robot unable to lift heavier objects, requiring its policy to be adjusted "in the wild". This change impaired the learned policy and the agent's success rate on the instructions referring to a red square dropped to $0.43$. After fine-tuning the policy with a frozen discriminator, the success rate went up to anywhere between $0.53$ and $0.67$,

## 5 CONCLUSION

In this paper we proposed AGILE, an IRL-inspired approach to training an agent to perform instructions with examples of goal-states. Our experiments in controlled settings with verifiable reward show that AGILE delivers performance which is comparable to that of RL and achieves more than 50% success rate with $2^{13}$ examples, a number that is amenable to crowd-sourcing. An attractive property of our approach is that learning "what should be done" and "how it should be done" is performed by two different model components. Our experiments confirm that the "what" kind of knowledge generalizes stronger and can help the agent to adapt to unseen instructions and to changes in the environment. We hypothesize that regularizing the discriminator could help with data efficiency and will experiment in this direction in our future work.

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

# A    DETAILED RESULTS

## A.1    AGILE VS. RL

The training curves for all methods can be found in Figure 2a. AGILE typically started to learn earlier than RL-RP and proceeded at the same speed, but after reaching the optimum at roughly 130 million steps AGILE's performance started slowly deteriorating and dropped to $0.91$ success rate after 500 million steps (see the RL-RP and AGILE curves in Figure 2a). Notably, AGILE's success rate was always above pure RL.

## A.2    DATA EFFICIENCY

The performance of AGILE for datasets of different sizes can be found in Figure 2b.

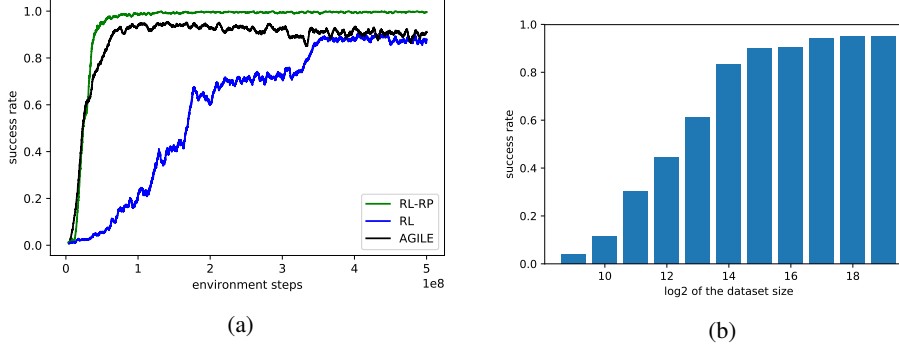

(a)                                                    (b)

Figure 2: (a) Learning curves for RL and AGILE task. RL-RP stands for RL with an auxiliary task of reward prediction. (b) Performance of AGILE for different sizes of the dataset of instructions and goal-states.

