# OpenReview forum: "Jointly Learning "What" and "How" from Instructions and Goal-States"
_ICLR.cc/2018/Workshop — Accept_

### Official Review · AnonReviewer3 · 2018-03-09

**Rating:** 6
**Confidence:** 3

**Review:**

This paper presents a framework (AGILE) for jointly learning a model for the reward (using a discriminator) and the policy. The task involves an agent performing operations in a simulated grid world, given an initial state and some instructions. Experiments show that AGILE out-performs vanilla RL, but falls short of the RL-RP method (RL with auxiliary reward prediction).

This paper is relatively well-presented. Although the task is simple and the results are not very strong, the model seems interesting, and I think this is good contribution for a Workshop paper.

Minor comment:
1) It would be nice to move Figure 1 earlier so the readers would more clear about the task or the input/output space.
2) The paper explains the model in a very high-level way. It would be helpful to provide a bit more detail (e.g. the architecture of the discriminator).

---

### Official Review · AnonReviewer1 · 2018-03-16
**The paper considers the problem of programming agents by providing statements in the form of relational statements. The proposed approach seems interesting, but a bit too basic and the experiments are too preliminary**

**Rating:** 7
**Confidence:** 4

**Review:**

Instead of providing a reward function or full demonstrations, this paper consider the cases where an agent's task is provided as a command describing a goal state in terms of spatial relations. Training is performed by providing couples (c,s), where c is a command and s is a desired goal state that corresponds to the command. The proposed algorithm, AGILE, trains a discriminator network to distinguish what should be a valid goal state from the reste. It also learns a policy that maximizes the frequency of visiting the goal states.
I think this work is overall based on a good idea. But I would be surprised if such an idea has not been extensively explored in the literature. The authors should clearly state what makes their work original. The experimental setup is too simple (5x5 grid) and the results are minimal.
Are the policy and the discriminator trained iteratively? It seems like you can just train the discriminator  to learn where to go, and then train the policy using the reward defined by the output of the discriminator. In that case, this problem is not different from standard binary classification and RL combined.

---

### Decision · Program_Chairs · 2018-03-20
**ICLR 2018 Workshop Acceptance Decision**

**Decision:**

Accept

**Comment:**

Congratulations, your paper was accepted to the ICLR workshop.